# Oxalate (dys)Metabolism: Person-to-Person Variability, Kidney and Cardiometabolic Toxicity

**DOI:** 10.3390/genes14091719

**Published:** 2023-08-29

**Authors:** Pedro Baltazar, Antonio Ferreira de Melo Junior, Nuno Moreira Fonseca, Miguel Brito Lança, Ana Faria, Catarina O. Sequeira, Luísa Teixeira-Santos, Emilia C. Monteiro, Luís Campos Pinheiro, Joaquim Calado, Cátia Sousa, Judit Morello, Sofia A. Pereira

**Affiliations:** 1Centro Hospitalar Universitário de Lisboa Central, E.P.E, 1150-199 Lisboa, Portugal; pedrombaltazar@gmail.com (P.B.); nuno.moreira.fonseca@nyu.edu (N.M.F.); miguellanca@campus.ul.pt (M.B.L.); luiscampospinheiro@gmail.com (L.C.P.); jcalado@nms.unl.pt (J.C.); 2iNOVA4Health, NOVA Medical School|Faculdade de Ciências Médicas, NMS|FCM, Universidade NOVA de Lisboa, 1150-082 Lisboa, Portugal; melo.junior@nms.unl.pt (A.F.d.M.J.); catarina.sequeira@nms.unl.pt (C.O.S.); luisa.santos@nms.unl.pt (L.T.-S.); emilia.monteiro@nms.unl.pt (E.C.M.); catia.moreirasousa@nms.unl.pt (C.S.); judit.morello@nms.unl.pt (J.M.); 3Centro Clínico Académico de Lisboa, 1159-056 Lisboa, Portugal; 4CHRC, NOVA Medical School|Faculdade de Ciências Médicas, NMS|FCM, Universidade NOVA de Lisboa, 1150-082 Lisboa, Portugal; ana.faria@nms.unl.pt; 5ToxOmics, NOVA Medical School|Faculdade de Ciências Médicas, NMS|FCM, Universidade NOVA de Lisboa, 1150-082 Lisboa, Portugal

**Keywords:** cardiovascular disease, hyperoxaluria, hypertension, metabolic disease, microbiota, nephrolithiasis, kidney disease, kidney stones, obstructive sleep apnea, systemic inflammation, pharmacology

## Abstract

Oxalate is a metabolic end-product whose systemic concentrations are highly variable among individuals. Genetic (primary hyperoxaluria) and non-genetic (e.g., diet, microbiota, renal and metabolic disease) reasons underlie elevated plasma concentrations and tissue accumulation of oxalate, which is toxic to the body. A classic example is the triad of primary hyperoxaluria, nephrolithiasis, and kidney injury. Lessons learned from this example suggest further investigation of other putative factors associated with oxalate dysmetabolism, namely the identification of precursors (glyoxylate, aromatic amino acids, glyoxal and vitamin C), the regulation of the endogenous pathways that produce oxalate, or the microbiota’s contribution to oxalate systemic availability. The association between secondary nephrolithiasis and cardiovascular and metabolic diseases (hypertension, type 2 diabetes, and obesity) inspired the authors to perform this comprehensive review about oxalate dysmetabolism and its relation to cardiometabolic toxicity. This perspective may offer something substantial that helps advance understanding of effective management and draws attention to the novel class of treatments available in clinical practice.

## 1. Introduction

Oxalate is the ionized form of oxalic acid, can reach systemic circulation through the diet or endogenous metabolism and is a metabolic end-product [1]. Metabolic alterations can lead to oxalate accumulation in body tissues, which is toxic. The kidney is a primary target of oxalate toxicity and its main excretory organ [2].

Calcium oxalate (CaOx) is the most common compound in kidney stones and is present in different forms, such as CaOx anhydrous, CaOx monohydrate, and CaOx trihydrate [3,4]. CaOx dihydrate is the most common crystal in healthy human urine, while CaOx monohydrate is the most common crystal in clinical stones as it is thermodynamically more stable [5].

Hyperoxaluria is an independent promoter of stone formation, as high oxalate concentrations or sharp CaOx monohydrate crystals can produce renal injury, enhancing CaOx crystal retention and stone formation [6]. Two distinct clinical expressions are defined for hyperoxaluria, namely primary hyperoxaluria (PH) and secondary hyperoxaluria. PH is caused by an inherited enzymatic deficiency in oxalate metabolism [7]. Secondary hyperoxaluria is mainly related to environmental factors (diet, lifestyle, and geographical, climatic, and ethnic factors (for a recent review, see [8])). Moreover, secondary hyperoxaluria has been described as a complex and multifactorial process and associated with metabolic diseases [9].

Nephrolithiasis is a relevant health issue that affects millions of people worldwide [10] and is expected to happen at some stage in life in up to 10 to 15% of the population in the developed world [11]. Nephrolithiasis is recognized as a burden on healthcare systems [12] due to its increasing prevalence, drug and surgical needs and frequent appearance as a recurrent disease. Described relapse rates vary among studies and can happen to more than half of affected individuals within 5–20 years [13]. 

Due to clinical association with nephrolithiasis [2], most evidence regarding oxalate dysmetabolism has been driven by the effects of oxalate in the formation of kidney stones and in kidney function. However, oxalate is also a driver of systemic inflammation [14], supporting an association between oxalate and disease beyond nephrolithiasis. Notably, oxalate has been associated with chronic diseases, including cancer [15], atherosclerosis, and arterial hypertension (HTN) [16]. To go further with this idea, we will assess the evidence regarding person-to-person variability in oxalate systemic levels and the genetic and non-genetic factors that predispose them to higher oxalate availability. Moreover, the mechanisms underlying oxalate toxicity will be uncovered to provide a rational and identify putative mechanistic linkers between oxalate dysmetabolism and cardiovascular and metabolic diseases.

## 2. Oxalate Metabolism, Person-to-Person Variability and Cardiometabolic Toxicity

### 2.1. Metabolic Pathways of Oxalate Production

The systemic availability of oxalate depends on dietary intake, intestinal absorption, endogenous production and renal excretion [17]. Moreover, the endogenous production of oxalate happens mainly in the liver [18,19] and accounts for around 90% of oxalate eliminated in urine [20].

Two main sources of oxalate production have been identified, namely ascorbic acid (vitamin C) and glyoxylate. Ascorbic acid is non-enzymatically broken down into oxalate [21] and its supplementation increases endogenous oxalate levels [22]. Ascorbic acid availability in humans and other primates [23] relies only on dietary intake and therefore data obtained from animal models should be cautiously interpreted. 

Glyoxylate is converted into oxalate by several oxidases and dehydrogenases, including glyoxylate oxidase and lactate dehydrogenase (LDH). Glyoxylate has several direct precursors, including glycolate and the amino acids glycine, serine and hydroxyproline. Glyoxylate is both a product and a precursor of glycolate.

Glycine is also both a precursor and a product of glyoxylate. Glycine is metabolized into glyoxylate by D-amino oxidase (DAO) and glutamate glyoxylate aminotransferase (GGAT). However, glycine’s contribution to oxalate production is very small [24]. Conversely, glyoxylate is metabolized into glycine by alanine-glyoxylate aminotransferase (AGT protein, *AGXT* gene). AGT transfers an amino group from alanine to glyoxylate, yielding pyruvate and glycine. There are two isoenzymes in mammals, encoded by two different genes *AGXT* aka *AGXT1* (2q37.3) and *AGXT2* (5p13.2). AGT1, at the peroxisome, is more relevant in oxalate metabolism, while the mitochondrial form AGT2 is less active but catalyzes multiple other aminotransferase reactions [25,26]. Importantly, glycine can be conjugated with microbiota-derived benzoic acid into hippuric acid. Hippuric acid is a protective metabolite since it increases the solubility of urinary oxalate [27].

Glycine and serine are interconverted by serine hydroxymethyltransferase (SHMT). Serine is further metabolized into hydroxypyruvate by AGT, and then into D-glycerate by hydroxypyruvate reductase (GRHPR).

Another precursor of glyoxylate is 4-hydroxyproline (ratio 4-hydroxyproline:3-hydroxyproline 100:1), originated from post-translational hydroxylation of collagen proline residues by prolyl hydroxylases. 4-Hydroxyproline is catabolized into glycine via *trans*-4-hydroxy-L-proline oxidase or to glyoxylate via 4-hydroxy-2-oxoglutarate aldolase (HOGA). Proline can also be obtained from diet.

Glycolate (hydroxyacetic acid) is both a precursor to and a product of glyoxylate metabolism. Glycolate is metabolized into glyoxylate by glycolate oxidase (GO). In the other sense, glyoxylate is converted into glycolate by glyoxylate reductase, an enzyme that has also named GRHPR [28].

Seminal works from the 1960s and 1970s have also proposed the aromatic amino acids (phenylalanine, tyrosine and tryptophan) as oxalate donors [24,29,30], but clinical validation remains to be ascertained.

Glyoxal has been also described as a precursor of oxalate (for a review, see [31]). Glyoxal has been postulated as a potential source of glycolate in erythrocytes via the glyoxalase system, using glutathione (GSH) as a cofactor [32]. The synthesis of glyoxal is minor in physiological conditions [32], but it might be relevant in diseases with increased levels of glyoxal that arise from the oxidation of carbohydrates, glycated proteins and lipids (e.g., diabetes) [33,34].

The putative involvement of mitochondrial kynurenine aminotransferase-III in glyoxylate-to-glycine conversion was also recently postulated and merits careful further examination [35]. 

Overall, production of oxalate might arise from different pathways (Figure 1), and several gaps still exist, including the contributions of several precursors, cofactors, enzymes, and transporters in different cells/tissues. 

In the following sections, we will describe, integrate, and discuss several lines of pre-clinical and clinical evidence on the mechanisms underlying the toxic effects associated with oxalate accumulation (Section 2.3) and the high person-to-person variability in oxalate systemic concentrations (Section 2.4).

### 2.2. Handling of Oxalate by the Organism

The gastrointestinal tract (GIT) absorbs and secretes oxalate, affecting net oxalate GIT handling (for review, see [36]). Although the stomach is involved in the absorption of oxalate, the small and large intestine appear to be most important [37,38,39]. Oxalate handling involves passive paracellular and active transcellular pathways, mainly via SLC26 gene family transporter proteins such as SLC26A3 and SLC26A6 [40,41]. Accordingly, paracellular transport entails the movement of oxalate between epithelial cells [39]. This transport is influenced by the existing transepithelial electrical and concentration gradients that affect the oxalate anion, in addition to the characteristics of tight junctions [39]. SLC26A6 is abundant in the small intestine, secreting oxalate into the lumen, while SLC26A3 in the colon facilitates oxalate absorption [41,42,43].

The majority of oxalate present in the body (endogenous or exogenous) is excreted by the kidney [44,45]. The renal excretion of oxalate mainly occurs via glomerular filtration and tubular secretion, with the proximal tubule being a region of significant importance in this process due to its high expression of SLC26 transporters, specifically SLC26A6 and SLC26A1 [46]. This family of transporters facilitates the transfer of a group of anions, including Cl^−^, HCO_3_^−^, and sulfate, playing a crucial role in oxalate handling [47,48].

Concentrations above 300–500 µM might be reached at the cortical collecting duct and even higher concentrations can be found in PH or after bowel resection as reviewed by John Asplin in 2002, evidencing genetic and non-genetic factors underlying variable oxalate handling among individuals [49].

Jiang and co-workers (2018) reported that patients with recurrent kidney stones have higher expression of SLC26A6 in the kidney and increased renal excretion of oxalate compared to controls, which appears to contribute to the formation of kidney stones [50]. Moreover, in Slc26a6−/− mice, the excretion of oxalate was reduced by more than 50% compared to wild-type mice [51]. Despite numerous efforts to comprehend the mechanisms governing oxalate absorption and excretion processes, there remain gaps that need to be addressed (for review, see [52]). Understanding oxalate-handling mechanisms is crucial, as urinary oxalate excretion plays a role in kidney stone formation and may represent an important target in the pharmacological treatment of kidney stones.

### 2.3. Molecular Mechanims of Oxalate Toxicity

Adverse outcome pathways (AOPs) of oxalate have been described [53]. This evidence has been gathered at the tissue level and in vitro, including renal tubular epithelial cells, fibroblasts, and monocytes/macrophages, as follows. Among the mechanisms identified include oxidative stress, mitochondrial dysfunction, inflammation, endoplasmic reticulum stress, autophagy activation, cellular senescence and tissue remodeling.

#### 2.3.1. Oxidative Stress

Oxidative stress has been strongly associated with oxalate toxicity. For instance, a rat model of hyperoxaluria and crystalluria induced by ethylene glycol plus 1,25-dihydroxycholecalciferol showed OS manifestations, namely alterations in mitochondrial and cytosolic GSH, increased activity of antioxidant enzymes, and decreased of mitochondrial cytochrome c [54]. Moreover, exposure to oxalate in mice resulted in a downregulation of markers, including NAD(P)H quinone dehydrogenase 1 (NQO1) and superoxide dismutase 1 (SOD1), GSH, glutathione peroxidase (GSH-Px) and catalase (CAT), and increased malondialdehyde (MDA) in kidney tissue [55]. The exposure of HK-2 cells, a proximal tubular cell line, to potassium oxalate (5 mM) also decreased the expression of antioxidant genes *NQO1* and *SOD1* and increased the production of hydrogen peroxide (H_2_O_2_). In NRK-52E cells from rat proximal tubular epithelium, oxalate exposure (0–1 mM) promoted mitochondrial changes and an imbalance of the NADPH oxidase subunits Nox4/1 (protective role) vs. Nox2 (kidney injury) [56], which is known to underlie the pathophysiology of hyperoxaluria due to significant production of reactive oxygen species (ROS) [57]. Additionally, oxidative stress was also promoted by CaOx in THP-1 cells, a monocyte cell line, decreasing manganese superoxide dismutase activity and the reduced-to-oxidized glutathione ratio [58]. 

Wang and colleagues (2021) reported that the up-regulation of the nuclear factor erythroid 2-related factor/heme oxygenase-1 (Nrf2/HO-1) pathway by circadian clock gene brain and muscle ARNT-like 1 (BMAL-1) attenuated urinary CaOx stones formation. The up regulation of this pathway interfered with oxidative stress both in vitro (HK2 cells) and in vivo (animal model of CaOx stone). The authors also found out that *BMAL-1* polymorphysms were associated with CaOx stones by analyzing blood samples from kidney stone patients [59]. 

As stated in Section 2.1, the erythrocytes also contributed to the pool of oxalate and glycolate [32]. Ma and co-workers (2014) reported that oxalate-mediated oxidative stress in erythrocytes might contribute to renal tubular damage and stone accumulation in patients with hyperoxaluria [60]. 

Summing up, oxidative stress underlies oxalate AOPs, which include players as small molecules (e.g., GSH), enzymes (e.g., NADPH oxidases, SOD1, CAT) and transcription factors (e.g., Nrf2 and BMAL-1).

#### 2.3.2. Mitochondrial Dysfunction

Mitochondrial dysfunction has also been associated with hyperoxaluria [61]. For instance, Cao and co-workers (2004) demonstrated that oxalate decreased mitochondria membrane potential in the Madin–Darby canine kidney (MDCK) cell line and inhibition of phospholipase A2 attenuated this effect. Moreover, oxalate induced ROS formation in isolated kidney mitochondria [62]. In line with this, Zhang and colleagues (2017) evaluated the effect of MitoTEMPO, a mitochondria-target antioxidant, in preventing cell injury by inhibiting mitochondrial dysfunction and modulating oxidative stress in NRK-52E cells treated with oxalate (700 µM). MitoTEMPO attenuated oxalate-induced cell injury by decreasing LDH leakage, decreased lipid peroxidation and mitochondrial ROS (but not intracellular ROS) and ameliorated the disruption of mitochondrial membrane potential. MitoTEMPO also increased ATP levels and selectively modulated NADPH oxidase subunits, particularly increased the expression of Nox4 and p22 without affecting Nox2. The authors also reported that MitoTEMPO reduced the levels of osteopontin (OPN) and interleukin (IL)-6 induced by oxalate [63].

Another research group showed that oxygen consumption rate was decreased in CaOx-treated THP-1 cells. This result was also validated in primary monocytes from healthy subjects treated with CaOx [58].

Oxalate (0.05 mM) exposure in differentiated THP-1 macrophages was also associated with altered cellular bioenergetics, mitochondrial gene expression and complex activity, ATP levels, redox homeostasis, anti-bacterial response and cytokine secretion [64]. In clinical samples, a pilot study involving CaOx stones formers and healthy subjects demonstrated that monocytes, but not lymphocytes and platelets, isolated from blood samples had a decreased mitochondrial function [65].

#### 2.3.3. Inflammation

The contribution of oxalate to inflammation has been strongly supported by in vitro and in vivo studies (for a review, see [53]). For instance, in vitro models of renal epithelial cells, including the MDCK cell line and human proximal tubular epithelial cells (RPTEC), were used to demonstrate that oxalate exposure (0.35 mM) promotes inflammation by increasing cyclooxygenase 2 (COX-2) and facilitating NF-κB activation via the degradation of IκB-α, respectively [66]. The internalization of CaOx by rat tubular epithelial cells induced oxidative stress and tubular cell necrosis and ended up releasing ATP, which is a NLRP3 inflammasome agonist and consequently triggers NLRP3 activation and contributes to renal inflammation. These effects were dependent on MyD88, a central signaling adaptor in translating innate pattern recognition into NF-κB activation. In addition, the blockage of IL-1β prevented CaOx-induced kidney injury [67].

A mice model of progressive oxalate nephropathy obtained from a high soluble oxalate diet (i.e., high in oxalate with an absence of dietary calcium) showed higher deposition of CaOx in kidney interstitium and infiltrates of mononuclear cells, particularly monocytes/macrophages, when compared to control mice (i.e., high-oxalate diet in the presence of dietary calcium) [68]. Moreover, the study group had a higher kidney expression of *Nalp3* (same as NLRP3, a component of the inflammasome) and *Il-1β. Nalp3* deficiency (Nalp3-null mouse) seemed to be protective for oxalate toxicity, namely renal failure, cell death and infiltration of monocytes/macrophages in the kidney, without affecting oxalate metabolism, initial renal crystal deposition, and without elimination of the inflammatory process [68]. Inflammasome activation by CaOx also increased IL-1β secretion via the NLRP3/ASC/caspase-1 pathway in dendritic cells derived from bone marrow [67]. In vitro, CaOx (2.5 mM) was involved in the differentiation of human monocytes (THP-1 cell line) into M1 macrophages, an inflammatory phenotype [69]. Moreover, oxalate has also been associated with necroptosis, a regulated inflammatory mode of cell death [70,71]. For instance, CaOx induced necroptosis in vitro and in vivo, via Tumor Necrosis Factor—α (TNF-α)/TNF-α receptor/Receptor-Interacting Protein Kinase (RIPK) 1 and 3 and Mixed Lineage Kinase domain Like (MLKL) [70,71]. Interestingly, RIPK3 has been associated with NLRP3 inflammasome activation [72]. 

Recently, the treatment of proximal tubular cells LLC-PK1 with oxalate (0–0.5 mM) suggest the involvement of transient receptor potential vanilloid 1 (TRPV1) on cell damage in a dose- and time-dependent manner. The activation of TRPV1 was associated with an increase of arachidonate 12-lipoxygenase (ALOX-12) protein levels and, consequently, increased 12(S)-hydroxyeicosatetraenoic acid [12(S)-HETE], an endogenous TRPV1 ligand. Activation of TRPV1 led to the increase of intracellular calcium, which induced PKC-α subunit activation and, consequently, upregulation of Nox4 and ROS formation (superoxide and H_2_O_2_). These events triggered the activation of NLRP3 and the consequent secretion of IL-1β that mediated cell damage. Inhibition of TRPV1 and other downstream effectors attenuated the tubular cell inflammation induced by oxalate. These findings were confirmed in an in vivo model, namely rats fed with a hydroxy-L-proline diet. Finally, the inhibition of TRPV1 in vivo did not interfere with the degree of hyperoxaluria and urinary supersaturation, only with inflammation [73].

Most of the available data on the role of oxalate in inflammation arise from pre-clinical studies. However, there are a few studies in humans. For instance, a pilot study involving CaOx stones formers and healthy subjects reported higher levels of IL-6 in stones formers [65]. In agreement with these results, another study found that healthy participants who consumed a low-oxalate diet for 3 days before drinking a blended preparation of fruits and vegetables containing a large amount of oxalate (equivalent to the amount found in a spinach salad) presented increased plasma levels of IL-6 and decreased levels of IL-10, IL-13 and MCP-1 [74]. Overall, this study demonstrated that dietary oxalate content interferes with monocyte bioenergetics and immune response but highlights the person-to-person variability that exists in these responses [74].

Overall, the activation of NF-κB and NLRP3/ASC/caspase 1/IL-1β have been described as the main pathways underlying inflammation in AOP of oxalate.

#### 2.3.4. Endoplasmic Reticulum Stress

Oxalate toxicity has also been associated with endoplasmic reticulum (ER) stress. In a rat kidney fibroblast-derived cell line (NRK49F), oxalate (up to 2 mM) induced apoptosis mediated by ER stress. An upregulation of key proteins, including GRP78, CHOP, PERK, IRE1, and ATF6 [75] was reported in male rats fed with 5% potassium oxalate for 7 weeks. Oxidative stress was mitigated in the presence of N-Acetylcysteine, a known antioxidant and a GSH precursor, but a modest effect was observed in ER stress activation [75]. Accordingly, Bhardwaj and co-workers (2017) demonstrated that, in animal models of hyperoxaluria, OPN levels, ER stress (increases in GRP78, ATF4 and XBP1), and apoptosis were correlated with the extent of CaOx crystal deposition in the renal tissue [75,76]. 

#### 2.3.5. Autophagy Activation

Another mechanism that has been associated with oxalate toxicity is autophagy. In the NRK-52E cell line, oxalate-induced autophagy activation in a time- and concentration-dependent manner (five concentrations ranging between 0 and 1 mM). This was assessed based on the expression of the hallmarks Beclin-1 and LC3B-II (increased) and SQSTM1/p62 (decreased) and the formation of autophagosomes and autolysosomes (increased) [77]. Autophagy activation decreased cell viability, increased both ROS and LDH release, and decreased GSH and mitochondrial membrane potential. In addition, the exposure of NRK-52E cells to oxalate favored the adherence of CaOx crystals to the cells. According to the authors, these effects were in part mediated by oxalate-induced p38 phosphorylation [77]. The described alterations were confirmed both in vitro—using chloroquine, an autophagy inhibitor—and via knockdown of Beclin 1—and in vivo—in a rat model of hyperoxaluria induced by ethylene glycol [77]. Moreover, Song and colleagues (2021) reported that the excessive autophagy activation induced by oxalate, observed from an increase of Beclin 1 and LC3-II and a decrease of p62 proteins levels, led to ferroptosis, an iron-dependent cell death (confirmed by an increase in Fe^2+^ content and a decrease in glutathione peroxidase 4 protein expression), in HK-2 cells [78].

#### 2.3.6. Cellular Senescence 

Cellular senescence was also associated with high levels of oxalate [79]. The HK-2 cell line was exposed to different sources of oxalate for 72 h, including 24 h-urine samples from age- and sex-matched individuals with (18.6 ± 14.3 mg/day urinary oxalate) or without (4.0 ± 1.33 mg/day) CaOx-kidney stones, sodium oxalate and CaOx monohydrate. All stimuli increased β-galactosidase (SA-βgal) activity and protein carbonyl content (oxidative stress indicator) and decreased telomere length-regulated proteins mRNA (TRF1, TRF2 and POT1). However, p16INK4a was exclusive to sodium oxalate and kidney stones groups [79].

#### 2.3.7. Tissue Remodeling

Oxalate toxicity is also manifested in tissue remodeling, namely in the epithelial-to-mesenchymal transition process (EMT) and fibrosis. For instance, increased TGF-β I expression and deposition of collagen I and III, accompanied by increased excretion of oxalate and CaOx in urine, was reproducible in two models of hyperoxaluria, namely mice fed with hydroxy-L-proline, an oxalate precursor (Figure 1), and mice treated with ethylene glycol [80]. Moreover, the treatment of human proximal tubular epithelial HK2 cells with oxalate (0.5 mM) or CaOx induced morphological changes, namely from a cobblestone-like monolayer of epithelial cells to the dispersed and spindle-shaped cells with migratory protrusions. These changes were coincident with a decrease in epithelial (cytokeratin and E-cadherin) and an increase in mesenchymal markers (α-smooth muscle actin) and in TGF-β. Oxalate and CaOx also promoted the acquisition of migratory capacity in HK2 cells. These effects were blunted by treatment with or overexpression of bone morphogenic protein-7, which is an anti-fibrotic cytokine in the kidney [80]. These findings were also demonstrated in mice cells of the inner medullary collecting ducts, which are physiologically exposed to higher concentrations of oxalate [81].

Overall, multiple mechanisms for the oxalate toxicity have been identified in the compelling evidence resumed in Figure 2. 

The role of oxalate in oxidative stress and inflammation have been extensively described, which constitute interdependent processes that create a vicious cycle, perpetuating each other and associated with mitochondrial dysfunction, ER stress, autophagy activation, cellular senescence and tissue remodeling. Thus, inflammation and oxidative stress might be referred as a “common soil” for all abovementioned mechanisms and can also be the cause and/or consequence of oxalate toxicity. The knowledge of AOP of oxalate might allow the discovery of innovative pharmacological targets. For that, omics analysis can also provide valuable information about oxalate AOPs [82]. For instance, a transcriptomics study reported 2276 downregulated and 750 upregulated genes in HK2 kidney tubular cell line exposed to oxalate [82]. The same group showed, in a rat model of hyperoxaluria, differential renal expression in the pathways related to OS (including upregulation of NADPH oxidase subunits Nox2, Nox4 and P22phox, and Gpx2), KIM-1 (a marker of renal injury), crystallization modulators (upregulation of OPN, MGP, Fetuin B, CD44, Clusterin, Bikunin/AMBP) and to osteogenesis (upregulation of Runx1, Runx2, KRT18, KRT8, and THP downregulation) [83].

While mechanistic evidence still requires clinical validation, both in vitro and rodent studies have been showing that androgens increase the excretion of oxalate and renal CaOx crystal deposition, while estrogens go in the opposite direction. Implicated mechanisms include the androgen receptor regulation of hepatic glyoxylate oxidase [84], contribution of the HIF-1α/BNIP3 pathway [85], suppressed macrophages capability to phagocyte the crystals [86]; and the suppression of the NADPH oxidase subunit 2 (NOX2) and the oxalate induced oxidative stress by estrogen receptor β [87] (for a review see [88]).

Finally, another idea that arises from this section is the variability of the oxalate concentrations used in the different studies for a broad range of mechanisms. This highlights the need of defining the minimal toxic concentration of oxalate according to the mechanism involved. Therefore, to appropriately study the dose-effect relationship, a compilation of oxalate concentrations in different human populations and also of the variability that exists between people was considered useful and is presented in the next section. 

### 2.4. Person-to-Person Variability in Oxalate Concentrations

Several clinical studies have been dedicated to the quantification of oxalate levels in different populations. 

Oxalate levels in plasma arise from a contribution of genetic and non-genetic factors that dictate the intricate dynamics of supply, production, and excretion of the combined endogenous and exogenous oxalate loads in the body. For instance, a median plasma oxalate level of 9 μM (range 2.7–5.3 μM) [89] was quantified in 167 stable kidney transplant recipients, median age 52 (18–81) years old, 37% of whom were female and 38% had live kidney donors.

Another study found a median plasma oxalate concentrations of 46 μM (range 16–89 μM) in 135 non-PH dialysis patients, median age 62 (50–70.5) years old, 41% women, 42% European, 23% Surinamese. A negative correlation between residual diuresis and plasma oxalate was reported [90].

Another study included a total of 347 persons were divided in 4 groups: PH (N = 39), enteric hyperoxaluria (EH) (N = 151), urinary stone disease (USD) (N = 80) and control group (N = 77). The mean plasma oxalate levels (SD) were 6.7 (8.7); 5.5 (6.4); 2.1 (2.7); 2.0 (1.5) μM. Therefore, the levels in PH were 3 times higher than in controls or stone formers [91]. Notably, the variability and overlap within groups was high, highlighting the contribution of non-genetic factors [91].

Urinary oxalate tends to be higher in kidney stone formers than control individuals, even though the variability is high and considerable overlap between groups exists [91,92,93,94]. For instance, oxaluria was ascertained in a multicenter, prospective, observational cohort study aimed to identify risk factors for cardiovascular diseases (CVD), progression of chronic kidney disease (CKD), and mortality. A total of 3123 participants were enrolled, 46% caucasian, 45% female and mean age 59 ± 11 years old. Higher oxalate levels in urine were inversely associated with estimated glomerular filtration rate (eGFR) and with serum and urinary calcium. Urinary oxalate excretion was higher in men, in higher body mass index, diabetes, creatinuria and proteinuria. High levels of urinary oxalate were also found for persons taking renin angiotensin system inhibitors, thiazide diuretics and statins [95].

Another study found that 24-h urinary oxalate levels were elevated in 44% of 683 kidney transplant recipients (KTR); Caucasians, 43% female and mean age 51 ± 13 years old. Oxaluria was inversely associated with mortality due to infectious causes, age, smoking habits and markers of kidney function. Positive associations were found with urinary levels of phosphate and urea and no sex differences nor association with systolic blood pressure, lipids, triglycerides or diabetes mellitus were found [96].

Overall, a high person-to-person variability was found among these five studies. The following sections are dedicated to genetic and non-genetic causes that explain this variability.

### 2.5. Genetic Causes of Oxalate Dysmetabolism

#### 2.5.1. Mechanisms of Primary Hyperoxaluria

Lessons learned from PH brought knowledge about metabolic and toxic pathways of oxalate. PH is a family of three rare autosomal recessive disorders of the glyoxylate metabolism, causing hepatic overproduction of oxalate [97,98] and leading to crystallization within the renal tubule, nephrolithiasis and/or nephrocalcinosis.

PH is estimated to account for 1–2% of pediatric cases of End-Stage Renal Disease (ESRD), a condition that requires dialysis or organ transplantation in order to children to survive [97]. The reduction of the GFR causes elevation of plasma oxalate even before ESRD develops and systemic oxalosis ensues with significant morbidity, affecting bones, retina, skin, blood marrow, and vessels [97]. Measuring urinary oxalate in properly acidified samples is the cornerstone of any algorithm for the evaluation of PH, with an excretion in excess of 0.5–0.7 mmol (45–60 mg)/1.73 m^2^ per day being highly suggestive of PH [97]. As GFR decline progresses, urinary excretion becomes unreliable and plasma measurements are necessary. However, these are complex and not widely available, often requiring combinations of chromatographic and mass spectrometry techniques [99]. A definitive diagnosis of PH requires genetic testing, although for many years the gold standard diagnostic test had been the measurement from a liver biopsy [100] of the catalytic activities AGT [101,102] or GRHRP [103,104].

PH displays genetic (and allelic) as well as phenotypic heterogeneity. In PH1 (OMIM 259900), the most prevalent form, glyoxylate accumulates due to deficiency of the liver-specific peroxisomal AGT with further conversion to oxalate by LDH [101]. Mutations in the *AGXT* gene encoding AGT underlie PH1 [102]. The p.G170R variant accounts for ~30% of the mutated alleles identified [105]. This mutation, when in cis with the minor p.P11L/p.I340M haplotype, unmasks a mitochondrial target signal that misroutes AGT away from peroxisomes. In addition to p.G170R, p.G41R, p.F152I and p.I244T are also considered minor alleles requiring (MiR) variants. p.P11L and p.I340M are not mutations but, instead, highly prevalent benign polymorphisms, reaching a 1:3 prevalence in certain populations, which do not *per se* account for any phenotype. Mutations in the *GRHRP* gene (9p13.2) are responsible for PH2 (OMIM 260000) [103]. As for PH1, it is the accumulation of glyoxylate that causes oxalate production. The most prevalent alleles are the c.103delG and c.403_404 + 2delAAGT variants, adding up to 37% and 18%, respectively, of mutant alleles [105]. Finally, PH3 (OMIM 613616) is caused by mutations in the *HOGA1* gene, positioned in 10q24.2 [106]; the c.700 + 5G > T splice variant accounts for about 50% of all HOGA1 alleles [105]. Unlike PH1 or 2, it is not clear how a deficiency in the liver-specific mitochondrial enzyme HOGA encoded by *HOGA1* will cause hyperproduction of oxalate. As mentioned, PH1 is the most prevalent form of PH (80%), while PH2 and PH3 are each responsible for 10% of genetically characterized cases [97]. Based on preliminary whole exome sequencing data, a prevalence for PH of 1:58,000 and a carrier frequency of 1:70 was estimated for the overall population [105], a finding that contrasts with previous assumptions of an incidence rate of approximately 1 in 100,000 live births per year in Europe [7].

A few hyperoxaluric patients with no evidence of fat malabsorption will also fail to display variants in any of the above-mentioned genes. One alternative (candidate) gene is *SLC26A6*, expressed in intestinal and renal epithelia and coding for a bidirectional oxalate transporter. *Slc26a6-/-* mice have an increased net absorption of dietary oxalate, leading to hyperoxaluria and calcium oxalate lithiasis, highlighting the role of constitutive *SLC26A6* in limiting net intestinal absorption of oxalate [107]. Additional evidence came from whole exome sequencing research performed in a hyperoxaluric pedigree, in which the detected heterozygous p.R507W *SLC26A6* missense mutation co-segregated with the calcium oxalate nephrolithiasis [108].

The role of any of these genes for the heritability in the more common complex/multifactorial trait of nephrolithiasis is not yet established. Genome wide association studies have failed to identify susceptibility loci around any of the mendelian genes responsible for PH [109], although a small case-control study, enrolling 225 patients, did find an association between the *SLC26A6* p.G539R allele and lithiasis [110].

Genetic information is crucial for PH, since (i) it is the gold-standard for diagnosis; (ii) there are prognostic geno-phenotype correlations; (iii) genotype may anticipate therapeutic choices including pyridoxine (VB6) and RNA interference (RNAi) (see below). It has been established that renal survival (dialysis-free) is worse for PH1 compared to all others. In patients at the age of 40, renal survival is 43%, 82%, and 96% for PH1, PH2, and PH3, respectively [105]. For PH1 it was also found that those individuals harboring *AGXT* MiR variants not only had a later onset of symptoms but also progressed more slowly to ESRD [105]. Current guidelines recommend testing pyridoxine (VB6) responsiveness in all genetically confirmed PH1 [111], although it is expected that full response will mainly occur with the *AGXT* MiR variants, in particular the p.G170R and p.F152I mutations. Pyridoxine (VB6), a AGT cofactor, can partially rescue mistargeting by stabilizing the protein. 

#### 2.5.2. Treatment of Primary Hyperoxaluria

The primary focus of managing PH is to decrease the excretion of urinary CaOx in order to extend kidney function, delay the progression to ESKD, and minimize associated complications.

The standard treatment for ESRD involves implementing strategies to reduce three key factors: (1) the saturation of urinary CaOx, (2) the crystallization of CaOx, and (3) the production of oxalate. To manage the risk of saturation, the most important approach is increasing fluid intake to achieve a high urinary output of more than 3 L/day. Additionally, dietary restriction of oxalate has a minimal impact on PH patients since the majority of oxalate is endogenously produced in these cases. To mitigate the risk of crystallization, oral potassium citrate is administered as it forms complexes with calcium, preventing the formation of CaOx crystals.

In the past, the reduction of endogenous oxalate production in PH has primarily been achieved via the administration of pyridoxine or by resorting to liver transplantation. Current guidelines recommend testing pyridoxine (VB6) responsiveness in all genetically confirmed PH1 (see above). However, it is important to note that most PH1 patients, as well as those with PH2 and PH3, will not have pyridoxine-responsive mutations. Supportive measures are often sufficient in reducing disease burden for PH3 patients [112].

In cases of PH1, liver transplantation may be necessary to restore the deficient enzyme and normalize oxalate levels. This intervention is commonly considered for patients with ESRD, and it often involves combined liver and kidney transplantation to effectively address the condition. However, there are emerging pharmacological approaches that aim to reduce the endogenous production of oxalate. These innovative therapies show significant promise in alleviating the burden of PH and may potentially decrease the need for more invasive procedures such as organ transplantation. By effectively managing oxalate production, these treatments have the potential to improve long-term outcomes and enhance the quality of life for individuals living with PH.

Currently, there are two potential targets for the treatment of PH: glycolate oxidase (GO) and the major hepatic form of LDH. These targets can be addressed using either enzyme active site inhibition or RNAi.

RNAi is a natural process that involves the suppression of gene expression using small interfering RNA (siRNA). siRNA is produced from double-stranded RNA (dsRNA) and is cleaved by an enzyme called ribonuclease III. When the siRNA binds to a cytoplasmic protein complex known as the RNA-induced silencing complex, it prevents the translation of the corresponding protein. Since its discovery in 2004, RNAi technology has been extensively utilized in drug development and has advanced into clinical trials [113].

Lumasiran, a liver-directed chemically modified short oligonucleotide complementary to the glycolate oxidase transcript, is an RNAi therapy that received FDA and EMA approval in 2020 for the treatment of PH1 [114,115]. Studies conducted on GO-deficient (*Hao1* knockout) mouse models have demonstrated that elevated levels of glycolate do not cause any adverse effects. Similarly, rare instances of individuals with biallelic loss of function variants in the *HAO1* gene have shown high levels of serum and urine glycolate without experiencing any disease-related symptoms [116,117].

In a recent double-blind, multinational, placebo-controlled, randomized controlled trial involving both adult and pediatric patients with PH1, lumasiran exhibited a substantial decrease in 24-h urine oxalate levels. The treatment resulted in a significant reduction of 53.5% (with a 95% confidence interval ranging from 62.3% to 44.8%), whereas the placebo group experienced a mean reduction of 11.8%. Importantly, no serious adverse events were reported during the course of the trial [115].

There are three ongoing phase III trials evaluating the efficacy and safety of lumasiran in patients with PH1, with expected completion dates in 2024 and 2025. The short-term nature of available data raises questions about the long-term effectiveness of lumasiran in treating PH1. The BONAPH1DE study is a prospective observational study that aims to evaluate the natural course of PH1 and the long-term efficacy and safety of lumasiran [118].

Inhibition of the enzyme LDH, which is responsible for converting glyoxylate to oxalate, has shown effectiveness in treating PH1 and PH2 in mouse models. Reduction of LDH expression via RNAi targeting *LDHA* transcript has resulted in a decrease in serum oxalate levels and the prevention of renal CaOx crystal deposition [119,120]. However, there are concerns regarding the use of RNAi to target LDH due to its significant role in glycolysis and the Cori cycle. In individuals with LDHA deficiency, engaging in high-intensity activity can lead to muscle weakness and breakdown. Studies have indicated that suppressing hepatic LDHA expression via RNAi may have metabolic effects, including potential alterations to tricarboxylic acid (TCA) cycle metabolites [112]. These findings emphasize the importance of monitoring the consequences of these effects to ensure safety and efficacy. Nonetheless, in the case of healthy volunteers treated with nedosiran, no musculoskeletal events were reported, and their levels of plasma lactate, pyruvate, and creatine kinase remained similar to those observed in the placebo group [121].

Additionally, nedosiran, which is administered via monthly subcutaneous injections, has demonstrated sustained and long-term reduction in urinary oxalate levels for patients with PH1 and PH2, bringing them closer to the normal range. While the precise mechanisms of oxalate production in PH3 are not yet well understood, it is plausible that silencing LDH could also be beneficial in this condition. Therefore, clinical trials involving nedosiran will be expanded to include PH3 patients as well as PH1/2 patients with ESKD (NCT04555486 and NCT04580420) [122].

In cases where RNAi therapy may not be feasible or accessible, alternative agents for the treatment of PH1 have been investigated. Stiripentol, initially developed as an antiepileptic drug for Dravet’s syndrome, has been investigated for its potential in treating PH1. It exhibits strong inhibitory effects on LDH and has shown the ability to reduce urine oxalate levels in cell cultures and animal models. However, when administered to PH1 patients who are dependent on dialysis, stiripentol has demonstrated limited effectiveness in lowering plasma oxalate levels. This suggests that stiripentol may not be a suitable treatment option for all types of PH1 patients, particularly those with a severe phenotype [123,124].

Another area of research focuses on exploring the potential of *Oxalobacter formigenes*, a bacterium, for actively reducing oxalate levels in patients with PH. It has been observed that Oxalobacter secretes a substance that enhances the secretion of oxalate in the intestine, offering a potential extrarenal mechanism for eliminating oxalate from the bloodstream. However, clinical trials examining this approach have produced inconsistent and primarily negative results [125,126,127].

Another therapeutic possibility in the field of PH1 could be a targeted gene utilizing CRISPR/Cas9 technology with human-induced pluripotent stem cells. A proof-of-concept study has demonstrated the effectiveness of CRISPR/Cas9-mediated integration of an *AGXT* minigene into patient-specific iPSCs, offering a promising strategy for generating functional hepatocytes that could potentially be used in autologous cell-based gene therapy for PH1 treatment. Additionally, CRISPR/Cas9 technology could be employed in PH1 by inhibiting the LDH or GO. However, available clinical data regarding its effectiveness is lacking [128]. 

Recent developments have focused on the gut and the potential of oxalate-degrading bacteria to control the intestinal absorption of oxalate in addition to oxalate production. In fact, intestinal secretion is an alternative to oxalate excretion in ESKD or other disease phenotypes characterized by high levels of bioavailable oxalate [38]. Recently, the role of intestinal bacteria in controlling intestinal oxalate flux in humans has received a great deal of attention among the non-genetic factors related to oxalate metabolism. 

### 2.6. Non-Genetic Factors in Oxalate Metabolism 

In addition to sex differences, males have up to three times more incidence of CaOx nephrolithiasis than females [10]; body mass, diabetes (see Section 2.3), and other non-genetic factors have been described. In fact, factors associated with oxaluria and oxalemia have been ascertained in different populations; a few among the more recent follows. 

#### 2.6.1. Kidney Disease

Oxalate undergoes glomerular filtration and concentrations up to 0.005 µM might be physiologically achieved and increase up to 20-fold in persons with CKD [67,68,70,71,129,130]. 

#### 2.6.2. The Microbiota

The microbiota comprises an enormous collection of microorganisms (viruses, bacteria, archea and eukaryotes) and the majority reside in the GIT. The result is a unique environment with a characteristic metabolism, complementary to the host, constituting a complex ecosystem.

A disrupted microbiota (a dysbiotic microbiota, with either reduced commensal bacteria, increase of pathogens, or an overall reduction in diversity and richness [131]) is associated with poor health outcomes. Although it is fairly resilient, the specific compositional features differ among individuals, and it can be altered by both internal and external stimuli. This microbiota plasticity, along with interindividual differences, makes it difficult to determine a “healthy” microbiota, but it also brings the opportunity to use this plasticity to shape the microbiota. By manipulating external factors, the architecture and biological outputs of the microbiota can be orchestrated to improve human health. Relations between a dysbiotic gut microbiota and disease have been established in different physiologic systems, including the gut–kidney axis. It has also been explored and related to CKD acute kidney inflammation/injury and nephrolithiasis [132].

Several metabolic diseases such as obesity or diabetes have a degree of dysbiotic microbiota associated and are also risk factors for lithiasis [133,134,135,136]. Lithiasis has been related with the gut microbiota, mainly due to calcium and oxalate metabolisms. Oxalate is present in food, particularly fruit, vegetables, and nuts, and is absorbed as a dietary component. The absorption rate is a function of the oxalate/calcium ratio [137,138]. Intestinal bacteria are able to degrade oxalate and to regulate oxalate transport in the intestinal epithelial. *Oxalobacter formigenes* colonizes the colon and uses oxalate to produce energy, playing a key role in oxalate homeostasis. Oxalobacter spp., Bifidobacterium spp. and Lactobacillus spp. [139,140,141] have been described as having the ability to degrade oxalate into formate via formil-CoA transferase and oxalyl-CoA decarboxylase [142,143]. The resultant formate is secreted back into the gut [40]. Therefore, the absence of these bacteria induces hyperoxaluria [144]. 

The bacteria or their metabolites interfere with transporter activity and expression, reinforcing the role of the microbiota in oxalate availability [145]. For instances, studies in vitro have demonstrated that *Oxalobacter formigenes*-conditioned media stimulate oxalate secretion via SLC26A6 [145]. The secretion of oxalate back into the intestine is mediated by the transporter SLC26A6, limiting its availability [40,41,146,147]. In addition, oxalate can be absorbed via paracellular transport. 

Indeed, the link between microbiota and lithiasis has been highlighted when authors found an association between antibiotic exposure of children at young age (3–12 months) and nephrolithiasis [148]. The study associated five different antibiotics with increased odds of nephrolithiasis [148]. Knowing the impact of antibiotics on the microbiota, the relation between these two conditions has been reinforced. The causality of this relationship has yet to be proven, but this information increases awareness of antibiotic generalization risk. Strategies to modulate the gut microbiota have not been sufficiently explored in kidney disease, but could be a mitigating approach for this situation, with a possible impact on lithiasis risk. The use of pre- or probiotics which modulate microbiota in a more oxalate-degrading feature has been suggested as a direction to move forward [149].

Several studies have characterized gut microbiota in stone and non-stone-formers (reviewed in [150]). Gut microbiota have distinct characteristics between stone-formers and controls. Overall, stone-forming patients seem to have a reduced microbial diversity [151,152,153], and some bacterial strains correlate with urinary constituents. Also, short-chain fatty acid (SCFA)-producing bacteria were less abundant in gut microbiota of stone-formers [152,154]. 

Furthermore, not only can the gut microbiota predispose, initiate, or promote the stone formation process, but the urinary microbiota has also been thought to have a role in this process, although the specific mechanisms are not completely understood [155,156]. 

Urinary bacteria-derived mechanisms may include: interaction with CaOx crystals to facilitate crystal growth and aggregation; changes in the urine chemistry promoting a more lithogenic environment; modulation of stone matrix proteins and expression of pro-inflammatory factors; and macrophage polarization towards a more inflammatory profile M1, all of which have an impact on stone formation process [150]. In fact, *Escherichia coli* and urea-splitting organisms (such as *Proteus*, *Staphylococcus*, *Klebsiella*, and *Pseudomonas*) are considered to play a role in nephrolithiasis. *Escherichia coli* produce citrate lyase, leading to hypocitraturia and consequently CaOx supersaturation with stone formation. In their turn, urea-splitting organisms, particularly *Proteus mirabilis*, decompose urea in urine which generates products that alkalinize the urine and participate in the formation of infection stones [157,158,159].

A study aiming to determine dysbiosis associated with urolithiasis showed an antibiotic-drive shift in the microbiome from a protective profile to a less-protective profile. Enterobacteriaceae and Lactobacillus presence was distinctive in urine of stone-forming patients and controls, respectively [160]. The distinct constitution of the urinary microbiota was also observed in a cohort of stone-forming patients and healthy controls [153]. Moreover, the bacterial profile of urine samples has more similarities with stone bacterial profiles than with stool microbiota [160], suggesting a putative role of the urinary microbiota in stone formation.

There is a myriad of predisposing factors for kidney stone diseases, particularly lithiasis, and among them are the presence of underlying diseases such as obesity, HTN, diabetes, CVD, and metabolic syndrome. These non-communicable diseases have a close association with gut dysbiosis, via distinct bacteria profile and bacterial metabolite production, at different steps of the disease, from the onset to the progression, and therapeutics [161,162], but also at the prevention level. Different microbiota-modulating strategies have been used to ameliorate the metabolic profile in non-communicable diseases, and via this approach, a not-so-direct but relevant impact can also be obtained for risk of lithiasis [163].

Diet has been recognized as a major driver for microbiota changes as nutrients can promote or inhibit microorganisms’ growth, shaping the gut microbial community. Moreover, diet-derived antigens and compounds can shape the gut microbiota in an indirect fashion by affecting the host’s metabolism and their immune system [164]. 

In addition to prebiotics and probiotics, other strategies directed at microbiota modulation have been proposed. For instance, nutritional intervention for microbiota correction in the context of lithiasis has not been fully explored, since these approaches remain much focused on oxalate metabolism and intake. Different dietary patterns have been associated with the gut microbiota in individuals with and without kidney stones [165]. However, the dietary pattern of the non-stone-formers was not characterized. One could expect that a dietary pattern such as the Mediterranean diet, with consistent effects on microbiota modulation, would increase SCFA production and decrease inflammation [166] and maintain/increase species involved in oxalate metabolism, contributing to reducing the risk of lithiasis. Few studies have explored this issue and have always centered on bacteria with oxalate-degrading enzymes [167,168]. The results are inconsistent, which raise the need to take a broad approach to microbiota, not only directed toward the oxalate-degrading bacteria. Moreover, a more personalized rationale should be used for different individuals and lithiasis specificities. The fecal microbiota transplant, which may have a great therapeutic potential, only has indication to be used for *Clostridium difficile* infection. Some clinical trials for the treatment of metabolic diseases have highlighted the treatment’s potential [169,170], but the only attempt to use it for urinary stone disease has been made in an animal model [171,172].

The studies that focused on microbiota have mainly explored the microbial composition, and bacteria in particular. This is important but limited information. An effort should be made to go beyond this and explore the functionality of the microorganism’s community, making use of metagenomics, metatranscriptomics and metaproteomics techniques. Big data science is crucial to better understanding not only the composition but the function and productivity of the microbiome. Metabolomics will provide the evidence of the effective productivity of this community. Efforts have been made to uniformize microbiota study in the field [173], standardizing procedures in order to have more and better-quality data that will allow us to move knowledge forward in the field.

#### 2.6.3. Other Non-Genetic Factors in Regulation of Oxalate Metabolism

Studies have investigated the impact of hormones, non-alcoholic fatty liver disease (NAFLD) and Crohn’s disease on oxalate synthesis and excretion. For instance, the effect of glucagon on oxalate synthesis was investigated by Holmes and co-workers (1995) by injecting guinea pigs with this hormone. Glucagon increased mean urinary oxalate excretion without affecting its peroxisome synthesis, despite decreasing the levels of hepatic peroxisome enzymes involved in oxalate synthesis. Interestingly, glucagon decreased hepatic levels of alanine, lactate and pyruvate, but did not affect glycolate and glyoxylate levels [174]. 

Moreover, Gianmoena and colleagues (2021) identified alterations in glyoxylate metabolism using an integrated epigenome and transcriptome analysis of hepatocytes from an obese (ob/ob) mouse, a widely used model for NAFLD. The researchers observed a down-regulation of *AGXT* mRNA and hypermethylation at the cyclic AMP response element (CRE) of the *AGXT* promoter. This was accompanied by an increase in oxalate formation from hydroxyproline metabolism. These results were validated in a second mouse model of NAFLD, in which a mouse received a Western-type diet. The authors also isolated hepatocytes, which were non-steatotic, from an *Agxt*-knockout mouse to confirm that *Agtx* deficiency leads to the production of higher levels of oxalate from the three precursors, namely glyoxylate, hydroxyproline and glycolate. However, viral-mediated *Agtx* transfer or inhibition of the catabolism of hydroxyproline decreased oxalate release. The translational relevance of these findings was explored in primary human hepatocytes isolated from male steatotic and lean donors. Hypermethylation was also observed, as well as the decrease in *AGXT* mRNA levels in steatotic donors. Moreover, *AGXT* mRNA expression showed a significant inverse correlation with triacylglycerols (TGA) content, whereas the DNA methylation of *AGXT* promoter presented a positive association with TGA. Contrary to mouse data, DNA methylation affected the recognition sites of hepatic nuclear factor 4, transcription factor sp1 and transcription factor AP-2alpha. Like *AGXT*, *GRHPR* and *HOGA1* mRNA expression showed a significant inverse correlation with TGA content. These results suggest that alterations in glyoxylate metabolism were more complex in human NAFLD. Overall, this study pointed out a mechanistic explanation for the higher risk of kidney stone disease in NAFLD patients [175].

On the other hand, Zong and colleagues (2016) investigated the potential role of GRHRP in apoptosis of intestinal epithelial cells in a model of Crohn’s disease. For that, a TNBS-induced experimental colitis in vivo model was used. The authors showed that TNBS induced an increase in GRHPR protein levels in the cytoplasm of intestinal epithelial cells of the animals. Moreover, it was shown that biochemical markers of apoptosis, namely active caspase 3 and cleaved PARP, were increased by TNBS treatment. Thus, it might suggest the involvement of GRHRP in the apoptosis of intestinal epithelial cells. The in vivo results were corroborated by in vitro experiments using TNF-α stimulated HT-29 cell line and transfected HT-29 cell line with GRHPR siRNA. To further validate these results, human colonic biopsy specimens from Crohn’s disease patients were analyzed, showing a strong staining of GRHPR in intestinal epithelial cells when compared to normal controls [176].

Another layer of regulation of oxalate levels is its intestinal transport, either absorption or secretion. As mentioned before, SLC26A6 plays an important role in controlling the net absorption of ingested oxalate. Post-translational modifications regulate its activity, namely N-glycosylation at amino acids 167 and 172 within the putative second extracellular loop, particularly its efficacy on oxalate transport, not influencing its plasma membrane delivery [177]. This can be critical in patients with cystic fibrosis, in which levels of bacterial exoglycosidase activity are increased and promoted SCL26A6 deglycosylation, thus inhibiting intestinal secretion of oxalate [177]. Moreover, Amin and colleagues (2018) reported that *SLC26A6* mRNA and protein levels were decreased in the jejunum of obese ob/ob mice, which is likely mediated by obesity-associated systemic and intestinal inflammation. Using an in vitro approach, the authors demonstrated that TNF and IFN induced a reduction in *SLC26A6* mRNA and protein levels in human intestinal CaCo2—BBE cells [178]. On the other hand, intestinal oxalate absorption predominantly occurs passively via the paracellular pathway. Bashir and co-workers (2019) demonstrated that obese mice had enhanced gut passive paracellular permeability, which was associated with reduction in mRNA and protein levels of occludin, zonula occludens-1, claudins-1 and -3, resulting in a significantly increased of oxalate absorption. This alteration to intestinal permeability was likely mediated by obesity-associated systemic/intestinal inflammation and oxidative stress [179].

While few studies are dedicated to the topic, these promising results pave the way for novel paradigms in oxalate metabolism.

## 3. Oxalate as a Common Link in Metabolic Comorbidities

According to the *European Association of Urology Guidelines 2023*, the presence of metabolic diseases is considered a risk factor for the development of nephrolithiasis and for its recurrence, leading to the classification of patients as *high-risk stone-formers*. Importantly, there is a bidirectional association, and oxalate-induced nephrolithiasis also influences [180] CKD, coronary artery disease, HTN [181,182,183], atherosclerosis [16,143,144,184], type 2 diabetes mellitus [185], dyslipidemia, and obesity [183,186] and NAFLD [175,187].

Approximately two-thirds of the patients with nephrolithiasis have malabsorption-associated hyperoxaluria and, upon presentation, most of them develop HTN, diabetes, and/or a history of CKD [188]. Malabsorption-associated hyperoxaluria is related with fat malabsorption, which leads to the formation of dietary calcium-fatty acid complexes, which results in inadequate intestinal oxalate binding and subsequent increased oxalate absorption. It occurs, for example, due to pancreatic disorders, after partial gastrectomy, bariatric surgery, jejunoileal bypass, and inflammatory bowel disease [7,9,130,189,190,191,192]. 

The possible effect of the dietary load of oxalate and the balance of dietary CaOx on cardio-renal function and blood pressure is not straightforward and needs to be investigated in a population-based setting. A pioneering prospective study in an Asiatic population with a high prevalence and incidence of CVD reported that higher dietary oxalate increased the risk of developing CKD and HTN, which was aggravated by a low calcium intake. Epidemiologic findings also reported that hypertensive subjects excreted more oxalate in urine compared to normotensive subjects (34.8 vs. 26.5 mg/day) [193]. Curiously, *Slc26a6*-deficient mice had hyperreninemia and higher serum succinate levels, which contribute to blood pressure regulation by inducing renin release via the activation of a G-coupled protein receptor in the juxtaglomerular apparatus. The mechanistic links between SLC26A6 and HTN were also addressed by others [194]. 

Mechanistic data from experimental models provides evidence for the toxic effects of oxalate in vascular function [195,196,197]. Oxalate-enhanced release of proinflammatory cytokines [198] from M1 macrophages is known for its pivotal role in dyslipidemia and vascular inflammation [199]. Accordingly, peroxisome proliferator-activated receptor γ agonist medication has a proven ability to reduce urinary crystal deposition, oxidative stress, and inflammation in hyperoxaluric rats [200], and the benefit of the use of a sodium-glucose co-transporter-2 (SGLT2) inhibitor was proved in diabetic individuals and pre-clinical models [201]. The exact mechanisms underlying these effects need further research.

Moreover, CaOx stones are rich in cholesterol and phospholipids that induce the crystallization process [202]. This highlights the interplay between oxalate and lipid metabolism in the formation of kidney stones.

Mild-to-moderate hyperoxaluria is also frequently seen in obese stone-formers and a positive correlation between increased body size and elevated urinary oxalate excretion has been reported in population-based studies [203,204,205]. Although the mechanism(s) underlying this association, beyond dietary ingestion, remain(s) to be validated clinically, preclinical evidence has been bringing some clues to light. For instance, obese mice showed local and systemic inflammation, which contributed to reducing active transcellular oxalate secretion into the bowel via the anion exchanger *Slc26a6* and enhancing the gastrointestinal paracellular absorption of oxalate [178,179,186]. Moreover, obesity-associated cholinergic activity also leads to Slc26a6 inhibition [206]. Increased dietary ingestion of oxalate and alterations in intestinal microbiota may further contribute to obesity-associated hyperoxaluria [178,186]. In mice with obesity induced by leptin deficiency, Amin et al., 2018 [178] observed hyperoxaluria compared to lean control animals. 

In the context of diabetes, urinary excretion of oxalate is higher in individuals with diabetes mellitus [207]. Moreover, plasma levels of glyoxylate and glyoxal are higher in diabetic patients and possibly contributing factors to hyperoxaluria [207]. 

Urinary oxalate is also higher in CaOx stone-formers with metabolic syndrome compared to those without metabolic syndrome, and this increase is proportional to the number of components of the metabolic syndrome [178,179,186].

Recently, obstructive sleep apnea (OSA) has also been linked to nephrolithiasis. OSA is often associated with resistant HTN [208], loss of dipper profile of blood pressure [209], and increased risk of severe cardiometabolic dysfunction [210,211]. OSA plays a role in insulin resistance, HTN, dyslipidemia, hepatic steatosis, and atherosclerosis [212,213,214]. Similarly, intermittent hypoxia linked to OSA plays a major role in the pathophysiology of metabolic syndrome in terms of sympathetic activation, systemic inflammation, impaired glucose and lipid metabolism, and endothelial dysfunction [212]. According to Tsai and co-authors (2018), after adjusting for age, sex, and comorbidities, OSA patients have a 35% increased risk of nephrolithiasis compared to controls, though the relationship between OSA, body mass index (BMI), and kidney stones remains undefined, as does the effect of OSA on 24-h urine parameters [215,216,217].

Tallam and colleagues (2022) performed a retrospective study on a cohort of patients to compare urinary lithogenic risk factors in patients with kidney stone disease with or without OSA [216]. Patients with OSA were more commonly male, had higher BMI and had higher rates of HTN. When comparing 24-h urine testing results, patients with OSA had higher supersaturation of uric acid, higher titratable acid, and net acid excretion; lower urinary pH and supersaturation of calcium phosphate; and significantly higher levels of 24-h urinary, uric acid, sodium, potassium, phosphorous, chloride, sulfate and urinary oxalate (35 mg/day vs. 37 mg/day, *p* = 0.05). In this study, after accounting for BMI, OSA was independently associated with lower urine pH and urinary titratable acid.

Shahait and colleagues (2022) have also studied OSA patients with the intention of identifying whether OSA is independently associated with a distinctive set of 24-h urine abnormalities in a cohort of stone-forming patients. On univariate analysis, patients with OSA were older, with a higher BMI and higher likelihood of diabetes mellitus and HTN. Patients with OSA had higher 24-h total urine volume (2018 versus 1818 mL, *p* = 0.03), calcium (279.7 versus 208 mg, *p* = 0.02), and oxalate (41.6 versus 31.3 mg, *p* < 0.001), yet lower 24-h urine pH (5.75 versus 6.03, *p* = 0.001) [215].

Further studies are needed to determine whether varying degrees of OSA severity as well as differences between treatment-compliant and treatment-noncompliant patients alter the risk of nephrolithiasis and associated metabolic disease.

A recently identified player in the nephrolithiasis-metabolic disease-hyperoxaluria trio is the HIF-AhR axis [46]. Notably, the AhR pathway is activated in the kidneys of animals exposed to chronic intermittent hypoxia, which is a relevant model of OSA and its co- morbidities.

There is clear long-standing evidence from numerous studies that HIF-1α and AhR activities are interdependent [218,219,220,221,222,223].

In order to assess the impact of AhR activation on CaOx nephrocalcinosis in normoxic conditions, Yang’s group (2020) [224] used an animal model of mice intraperitoneally injected with glyoxylate to establish CaOx nephrocalcinosis, with or without a treatment with the AhR activator 6-formylindolo(3,2-b) carbazole (FICZ). This study revealed that the AhR-miR-142a-IRF1/HIF-1α axis attenuates CaOx nephrocalcinosis-mediated kidney injury and crystal deposition by modulating macrophage polarization (diminishing M1 and promoting M2). AhR expression was upregulated and negatively correlated with interferon-regulatory factor 1 (IRF1) and HIF-1α levels in a murine CaOx nephrocalcinosis model following administration of FICZ. Moreover, AhR activation suppressed IRF1 and HIF-1α levels and decreased M1 macrophage polarization in vitro.

Finally, oxalate has also been associated with the risk of sudden cardiac death in dialysis patients [225], cardiovascular risk in ESKD [226] and in a stone-forming population in rural China [227], and progression of CKD [95], defined as a predictor of kidney function in PH [228].

## 4. Conclusions

Oxalate metabolism is highly variable among individuals and higher concentrations have been associated with AOPs that ultimately promote kidney stones. So far, the AOPs associated with oxalate match five of the nine defined hallmarks of aging [229], including telomere attrition, cellular senescence, mitochondrial dysfunction, changed nutrient sensing and epigenetic alterations that underlie oxalate-related kidney and cardiovascular toxicity. Oxalate dysmetabolism has been associated with several cardiometabolic diseases, which can be related to common pathogenesis/AOPs and/or to common risk factors. Data from different studies need harmonization for a comprehensive integration of the evidence available in the literature (e.g., definition of cardiovascular risk, control for confounding factors, definition of optimal levels of oxalate, and the choice of doses to be used in pre-clinical models, should be considered when assessing translational potential). 

This heterogeneity among studies was considered a major issue in a meta-analysis published in 2020 [230] and a reason not to declare a substantial causal relationship between urinary stones and CVD. Kidney stones were associated with increased cardiovascular risk including coronary heart disease and stroke in a meta-analysis performed in 2014 [231]. In addition to kidney and CVD, NAFLD and OSA are also emerging topics to be further investigated.

Expanding evidence in the pathophysiology of PH is fostering the development of innovative drugs. Oxalate dysmetabolism in idiopathic stone formers is still to be unveiled, but its link to metabolic disease can give some clues. Moreover, more clinical data in models of cardiometabolic disease and a better understanding of the particularities of oxalate metabolism regulation in particular contexts (e.g., inflammation, hypoxia) and different precursors’ availability and metabolic shifts will shed light on this idea. 

This perspective may offer something substantial that helps advance understanding of effective management and draw forth a novel class of treatments available in clinical practice.

There are novel opportunities and promises in PH with innovative treatments (see Section 2.5.2) and for secondary hyperoxaluria by targeting endogenous precursors and endogenous production pathways (Section 2.1), targeting the absorption and secretion of oxalate in the gut (Section 2.2), development of nutritional interventions, microbiota modulators (Section 2.6.2), and new drugs to treat cardiometabolic disease (e.g., SGLT2i) (Section 3) and other putative novel targets such the inflammasome (Section 2.3.3), Nrf2 (Section 2.3.1), HIF or AhR pathway (Section 3).

## Figures and Tables

**Figure 1 genes-14-01719-f001:**
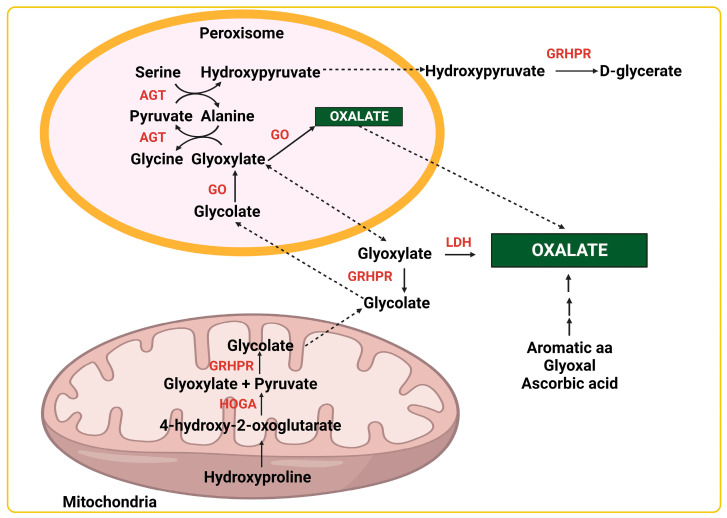
Pathways of endogenous production of oxalate. Primary hyperoxaluria types 1 and 2 are respectively associated with peroxisomal AGT and cytosolic GRHPR deficiency, resulting in accumulation of glyoxylate, which is a precursor of oxalate. Primary hyperoxaluria type 3 is caused by a defect in HOGA at the mitochondria. Other precursors have been described as contributors to bioavailable oxalate including the aromatic aa (tryptophan and phenylalanine, which are essential AA obtained from diet), ascorbic acid/vitamin C (from diet) and glyoxal (from carbohydrate and lipid oxidation in erythrocytes and the liver). The mechanisms leading to increased oxalate levels in secondary hyperoxaluria are not so well defined as for primary oxaluria. Glyoxylate is converted into oxalate by several oxidases and dehydrogenases, including glyoxylate oxidase and lactate dehydrogenase (LDH). AGT, alanine-glyoxylate aminotransferase; GO, glycolate oxidase; GRHPR, glyoxylate reductase–hydroxypyruvate reductase; HOGA, 4-hydroxy-2-oxoglutarate aldolase; LDH, lactate dehydrogenase; aa amino acids. Created with BioRender.com.

**Figure 2 genes-14-01719-f002:**
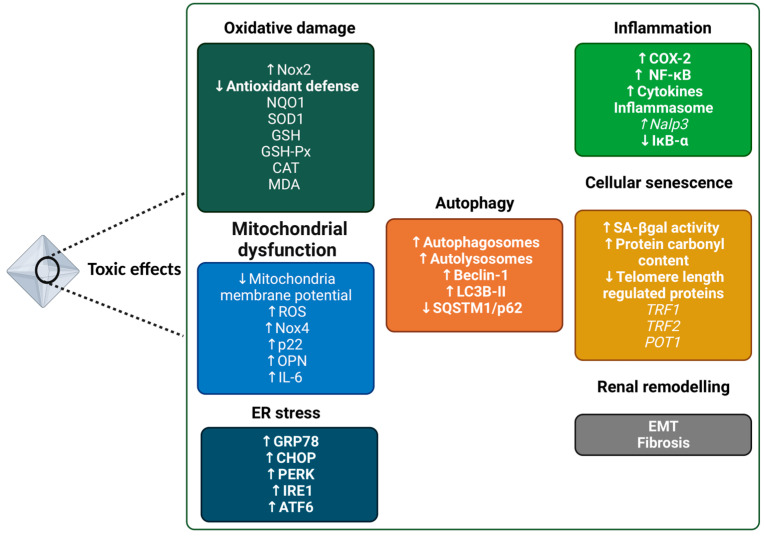
Mechanisms and mediators of oxalate toxicity. The mechanisms that have been unveiled encompass a range of factors, including oxidative stress, disruption of mitochondrial function, inflammatory responses, endoplasmic reticulum stress, autophagy activation, and alteration of tissue structure as demonstrated by several hallmark molecular changes herein shown. Nox2, NADPH oxidase subunit 2; NQO1, NAD(P)H quinone dehydrogenase; SOD1, superoxide dismutase 1; GSH, glutathione GSH-Px, gluthatione peroxidase; CAT, catalase; MDA, malondialdehyde; GRP78, 78-kDa glucose-regulated protein; PERK, PKR-like ER kinase; IRE1, inositol-requiring enzyme 1; ATF6, Transcription factor 6; CHOP, C/EBP homologous protein; COX2, cyclooxygenase-2; NF-κB, Nuclear factor kappa light chain enhancer of activated B cells; Nalp3, Nucleotide-binding oligomerization domain, Leucine rich Repeat and Pyrin domain containing-3; IκB-α, NF-κB inhibitor α; SQSTM1/p62, Sequestosome-1/ubiquitin-binding protein p62; SA-βgal, β-galactosidase; TRF1, Telomeric repeat-binding factor 1; TRF1, Telomeric repeat-binding factor 2; POT1, Protection of telomeres protein 1; EMT, epithelial-to-mesenchymal transition. ↑ increase; ↓ decrease.

## Data Availability

Data sharing not applicable.

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
