# Peer review of "Oxalate (dys)Metabolism: Person-to-Person Variability, Kidney and Cardiometabolic Toxicity"

_genes, 2023, doi:10.3390/genes14091719_

Round 1

Reviewer 1 Report

The authors have submitted a review article of illustrating a current knowledge regarding impact of variability of oxalate dysmetabolism followed by the levels of tissue oxalate on cardiovascular and metabolic diseases in humans. The authors searched a range of eligible literature, from well-known classical, and latest research regarding an association of the oxalate levels with cardiometabolic toxicity such as kidney stones, which are primarily attributed to the pathology of the oxalate-related diseases. The authors discussed the beneficial availability of a variety of oxalate metabolism in persons and tried to discuss the prediction of treatment for the cardiometabolic toxicity, resulting in reliable perspectives. This issue is of quite interest, and impact of their review is very strong. My overall concern with the review describing the current available data regarding beneficial availability of the person-to-person variability of oxalate metabolism is that information provided may offer something substantial that helps advance our understanding of effective management which draws novel class of effective treatments available in clinic. The reference list may be useful for readers who are interested in this issue.

To strengthen authors’ perspectives, the authors are strongly recommended to add a description what kind of disease treatment methodology can be considered for each cause that changes the oxalate level in persons, for instance. The opposite, toxicological effects of expected outcomes, if known, may influence largely the authors’ perspective for the oxalate dysmetabolism.

Reviewer 2 Report

Baltazar and colleagues have submitted an extraordinarily comprehensive overview of the current understanding of oxalate metabolism and toxicity.  They are to be commended for pulling together this very diffuse body of knowledge.

The data, for the most part, all seem to be present but this reviewer has several recommendations to improve the readability and understanding of this manuscript which are listed under broad headings.

1.  Adjustment of the language usage.

   There are run on sentences (lines 64-68), uncertain words (circutries, line 86), improper word usage (Endogenous metabolism, line 88, should probably be endogenous production), and ambiguous sentences (lines 93-94).  This reviewer recommends that the authors re-read and revise the language to improve understandability.

2.  Reorganization

  The authors seem to discuss a part of a topic, stop, then pick it up someplace else.  For example, under 2.2, Oxalate Toxicity, the authors state that they are going to talk about oxidative stress, ER stress, and inflammation.  They discuss oxidative stress first, then ER stress, then autophagy.  Throughout this discussion they go back to oxidative stress again.  It would be better to take one topic, discuss all aspects, then move to the next.

  The authors also need to make a distinction between mechanisms and mediators.  What is the mechanism for oxalate toxicity (e.g., oxidative stress) and what are the mediators?

  The focus of this article, as articulated by the title, should be the identification of factors that create differences in the response to oxalate generation.  The authors have a section entitled person to person variability in oxalate concentration, discuss genetics, and nongenetic factors.  However, they have also discussed sex difference in a previous section.  Why not include in the person to person differences?

3.  Removal of extraneous information

  The authors explore microbiota in great detail, the role in inflammation and stone formation, the predisposing risk factors for kidney stones, but the relevance to person to person oxalate metabolism is not really addressed.  This section could be shortened considerably without sacrificing the message that the authors wish to convey.

4.  The authors say little about renal handling of oxalate

5.  The subtitles throughout the manuscript do not really convey the contents of that section.  It would be best to divide 2.2 Oxalate toxicity into a discussion of mechanisms followed by a discussion of manifestations, or vice versa.  Section 2.5.2 is really about genetic and epigenetic regulation.  This is important but the title says Regulation of oxalate metabolism.

Additional comments

1.  Not sure what 2.1 Metabolic Circutries of oxalate means

2.  LInes 102-104 need a reference to back up that statement.

3.  Line 147, the authors need to make very clear whether oxalate toxicity is a cause of OS or a result of OS

4.  Line 225, not sure what the authors mean by a hyperfunction of TRPV1

5.  Figure 1.  I think the authors want to say Pathways of endogenous production of oxalate

The authors would improve the readability of this manuscript by revising the language considerably.

Round 2

Reviewer 1 Report

The authors have addressed properly all the issues raised by reviewers including me. I have no more comments, and now recommend that this manuscript is acceptable for publication in the journal Genes.